# LoFormer: Local Frequency Transformer for Image Deblurring

## ABSTRACT

Due to the computational complexity of self-attention (SA), prevalent techniques for image deblurring often resort to either adopting localized SA or employing coarse-grained global SA methods, both of which exhibit drawbacks such as compromising global modeling or lacking fine-grained correlation. In order to address this issue by effectively modeling long-range dependencies without sacrificing fine-grained details, we introduce a novel approach termed Local Frequency Transformer (LoFormer). Within each unit of LoFormer, we incorporate a Local Channel-wise SA in the frequency domain (Freq-LC) to simultaneously capture cross-covariance within low- and high-frequency local windows. These operations offer the advantage of (1) ensuring equitable learning opportunities for both coarse-grained structures and fine-grained details, and (2) exploring a broader range of representational properties compared to coarse-grained global SA methods. Additionally, we introduce an MLP Gating mechanism complementary to Freq-LC, which serves to filter out irrelevant features while enhancing global learning capabilities. Our experiments demonstrate that LoFormer significantly improves performance in the image deblurring task, achieving a PSNR of 34.09 dB on the GoPro dataset with 126G FLOPs. Code will be released.

## CCS CONCEPTS

• **Computing methodologies → Reconstruction**.

## KEYWORDS

self-attention, frequency domain, image deblurring

## 1 INTRODUCTION

The field of image deblurring has made significant advances riding on the wave of global feature learning methods. Some MLP-based methods have been proposed, *e.g.*, MAXIM [32] decomposes the global MLP operation into window-MLP and grid-MLP in a sparse manner (see Fig. 1(a)). In addition to MLP-based methods, recent research explorations [31, 34, 40] have shown the ability of Transformers in image deblurring task. Self-Attention (SA) [33], the key to capturing long-range dependency, has quadratic computational complexity *w.r.t.* the number of tokens, which is infeasible to be applied to high-resolution images in image deblurring. To make computation feasible, existing methods try various ways to reduce the number of tokens for SA in spatial domain, which can be categorized into three groups. (1) Local Spatial-wise SA (we use the

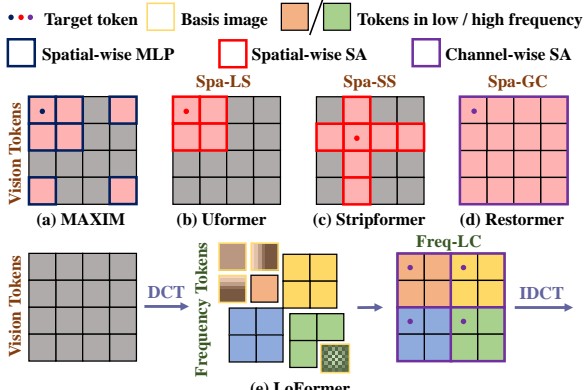

**Figure 1: Different architectures of global feature learning. (a) MLP method in MAXIM [32]; (b) Window Self-Attention in Uformer [34]; (c) Strip Self-Attention in Stripformer [31]; (d) Global Channel Self-Attention in Restormer [40]; (e) Local Frequency Self-Attention in LoFormer. The vision tokens in spatial domain are converted to the DCT coefficients (frequency tokens) of different DCT basis images.**

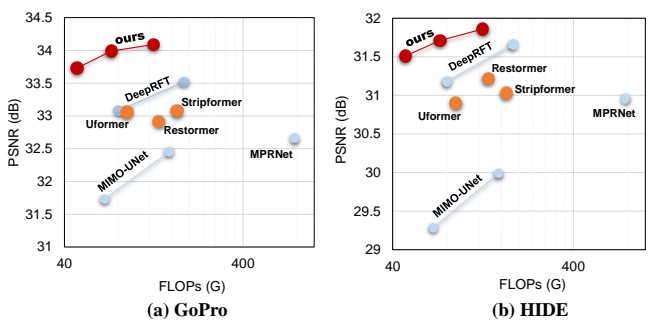

**Figure 2: PSNR vs. FLOPs on the GoPro and HIDE datasets. Our method performs much better than other state-of-the-arts, especially Transformer based methods highlighted in *orange*.**

abbreviation Spa-LS to represent **Spa**tial domain-**L**ocal **S**patial-wise SA). Uformer [34] proposes a local-enhanced window Transformer block to capture local context (see Fig. 1 (b)), which hurts long-range modeling. (2) Region-specific global SA. Stripformer [31] explores horizontal and vertical intra-strip and inter-strip SA (Spa-SS represents **Spa**tial domain-**S**trip **S**patial-wise SA) (see Fig. 1 (c)), which relies on a strong assumption that image blur is usually regionally directional. (3) Coarse-grained global SA. Restormer [40] captures long-range interactions via Global Channel-wise SA (Spa-GC represents **Spa**tial domain-**G**lobal **C**hannel-wise SA) (see Fig. 1 (d)). Though Spa-GC can be learned, it inevitably focuses more on extracting low-frequency components of the image due to two reasons: (i) the energy of the image mainly lies in low-frequency, and (ii) when

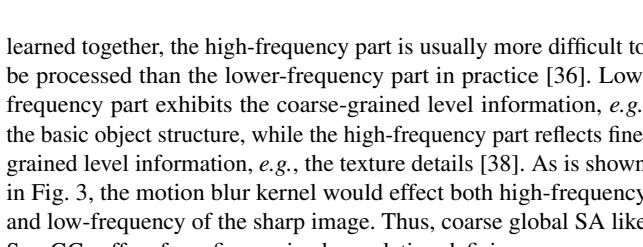

**Figure 3: The visualization of various generated kernels and their impact on the sharpness of images within both the spatial and frequency domains. Specifically, the first row of the visualization pertains to the degraded image, while the second row illustrates the generated kernel. Furthermore, the odd columns represent the spatial domain, while the even columns depict the frequency domain.**

learned together, the high-frequency part is usually more difficult to be processed than the lower-frequency part in practice [36]. Low-frequency part exhibits the coarse-grained level information, *e.g.*, the basic object structure, while the high-frequency part reflects fine-grained level information, *e.g.*, the texture details [38]. As is shown in Fig. 3, the motion blur kernel would effect both high-frequency and low-frequency of the sharp image. Thus, coarse global SA like Spa-GC suffers from fine-grained correlation deficiency.

To model long-range dependency without compromise of fine-grained details, we presents Local Frequency Transformer (Lo-Former) for image deblurring. Concretely, we propose **Freq**uency domain-**L**ocal **C**hannel-wise SA (Freq-LC) shown in Fig. 1 (e). First, we transform features into the frequency domain via Discrete Cosine Transform (DCT). DCT represents original features as the coefficients of different basis images. As shown in Fig. 1 (e), The basis images can be arranged in a rectangular grid, with lower frequency components in the top-left corner and higher frequency components towards the bottom-right. The top-left basis image represents the average intensity of the entire image, while the remaining basis images capture increasingly finer details and textures. The token at any frequency has global information. To allow equivalent learning opportunities for coarse-grained structures and fine-grained details, we design a window-based frequency feature extraction paradigm, *i.e.*, splitting frequency tokens into non-overlapping windows. The window on top left consists of tokens with coarse-grained structures (coarse tokens) and the one on bottom right consists of tokens with fine-grained details (fine tokens). Then, SAs are applied within local windows separately, capable of capturing cross-covariance within low- to high-frequency windows in parallel.

We further propose an intra-window MLP Gating (MGate) on the frequency axis complementary to Freq-LC, which performs a gating operation on the feature learned via SA. It's worth mentioning that the gating operation enhances the model capability of global information learning. We term our Freq-LC and intra-window MGate followed by a feed-forward network as **Lo**cal **F**requency **T**ransformer (LoFT) block, which is the basic building block of LoFormer.

The main contributions can be summarized as follows:

- We propose simple yet effective Freq-LC to model long-range dependency without compromise of fine-grained details

and introduce MGate which performs a gating operation and learns global features complementary to Freq-LC for better global information learning.
- We prove that Spa-GC equals to Freq-GC where coarse information dominates the calculation and verify that our Freq-LC has stronger capability in exploring divergent properties in frequency than Spa-GC.
- Extensive experiments show LoFormer achieves state-of-the-art results on image deblurring task, *e.g.*, 34.09 dB in PSNR for GoPro dataset. The PSNR (dB) *vs.* FLOPs (G) compared with state-of-the-arts are shown in Fig. 2.

## 2 RELATED WORKS

### 2.1 Deep Image Deblurring

Based on paired blurry-sharp image datasets, many methods [3, 4, 6, 13, 14, 18, 20, 30, 32, 35, 41] adopt an end-to-end strategy to train a deep neural network for image deblurring task. To achieve better performance, most of the improvements revolve around the network structure or the specific components. For example, MPR-Net [41] proposes a multi-stage architecture, which learns restoration functions progressively. MIMO-UNet [6] presents a multi-scale-input multi-scale-output UNet architecture to ease the difficulty of training. NAFNet [3] builds a network without activation and uses LayerNorm [1] (LN) to stabilize the training process with a high initial learning rate. Other methods such as Whang [35] introduces diffusion-based method for deblurring.

### 2.2 Low-level Vision Transformers

Transformer [33] was first proposed for natural language processing. Recently, several Transformer models are explored for low-level vision tasks, such as image denoising [2, 34, 40], deblurring [31, 34, 40], deraining [2, 34, 40], and super-resolution [15]. Like model from ViT [9], IPT [2] applies a pre-trained Transformer model based on ImageNet [8] dataset for various image restoration tasks. SwinIR [15] and Uformer [34] apply window-based SA [16] to capture long-range dependencies. Stripformer [31] decomposes the spatial-wise global SA into horizontal and vertical SA. Restormer [40] models global context by applying SA across channels with linear complexity rather than spatial. Though extensive

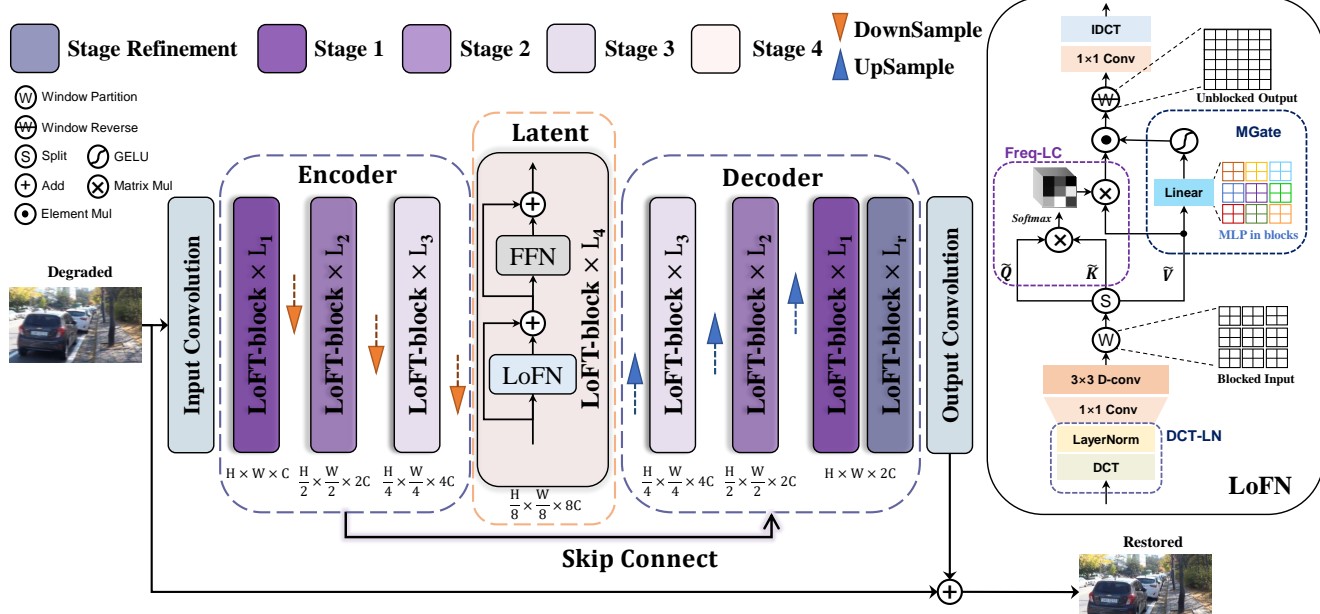

**Figure 4: Architecture of LoFormer. The main backbone of LoFormer is an UNet [27] model built in Restormer [40]. The basic building block of LoFormer is Local Frequency Transformer block (LoFT-block) , which consists of a Local Frequency Network (LoFN) module and an Feed-Forward Network (FFN) module. The core components of LoFN are DCT-LN, Freq-LC and MGate on frequency windows that perform global context aggregation.**

**Table 1: Hyper-parameters with LoFormer. $[L_1 \sim L_4]$ and $L_r$ denote the number of LoFT block in stage-1,2,3,4 and refinement. C is the feature dimension. PSNR(dB) is calculated on GoPro [20] dataset. FLOPs (G) are calculated on an input RGB image of size 256 × 256. Params are calculated in (M).**

| Model | $[L_1 \sim L_4]$ | $L_r$ | C | PSNR | Params | FLOPs |
|---|---|---|---|---|---|---|
| Restormer | [4,6,6,8] | 4 | 48 | 32.92 | 26.12 | 135 |
| LoFormer-S | [2,4,6,14] | 2 | 32 | 33.73 | 16.38 | 47 |
| LoFormer-B | [2,4,12,18] | 2 | 36 | 33.99 | 27.93 | 73 |
| LoFormer-L | [2,4,12,18] | 2 | 48 | **34.09** | 49.03 | 126 |

efforts have been made to capture long-range dependencies, either conducting pixel-wise SA on a local window or learning global context in a sparse manner, they neglect an important fact that independent components within an image/feature should not be blindly modeled altogether via SA operations. We propose to decompose features into independent components, *i.e.*, frequency tokens, by projecting them onto orthogonal bases via DCT and conduct global context learning within each partitioned frequency window.

## 2.3 Frequency Domain Applications

A growing literature corpus has proposed methods extracting information from the frequency domain to fulfill different tasks [26, 37, 45]. FcaNet [23] generalizes the channel attention in the frequency domain for image classification; GFNet [24] learns long-term spatial dependencies in the frequency domain. Zhong [44] proposes a novel

frequency enhancement module to detect camouflaged objects in the frequency domain. LaMa [29] uses the structure of fast fourier convolution [5] as the building block to image inpainting. Deep-RFT [18] introduces a simple res-fft-relu-block into deep networks for image deblurring. FourierUp [39] explores down/up-sampling operators in Fourier domain. DDCN [11] and CARD [19] make use of block discrete cosine transform to reduce compression artifacts. FADN [36] divides the input feature into multiple components based on a frequency-domain predictor for image super-resolution. Inspired by the success of frequency domain, we propose an LoFT block which enables (1) independent components decomposition via DCT, (2) globally independent context learning in a frequency window, and (3) low computation complexity.

## 3 METHOD

### 3.1 Main Backbone

An overview of LoFormer architecture is shown in Fig. 4. LoFormer employs a UNet [27] architecture proposed by Restormer [40] as the backbone. In Restormer, each stage of encoder-decoder contains multiple Transformer blocks. We design a **Lo**cal **F**requency **T**ransformer (LoFT) block as our building block. As illustrated in Table 1, we put more building blocks to lower stages for efficiency, *i.e.*, from stage-1 to stage-4, the number of LoFT blocks are [2, 4, 6, 14] (LoFormer-S), [2, 4, 12, 18] (LoFormer-B and LoFormer-L) and number of attention heads are [1, 2, 4, 8]. Besides, the stage-refinement contains two LoFT blocks.

**Figure 5: LayerNorm is added after DCT to force frequency tokens to be distributed equally.**

## 3.2 Local Frequency Transformer Block

LoFT block consists of a proposed Local Frequency Network (LoFN) and a Feed-Forward Network (FFN). As shown in Fig. 4, the LoFN consists of (i) LayerNorm after DCT (DCT-LN), (ii) Frequency domain-Local Channel-wise SA (Freq-LC), (iii) intra-window MLP Gating in the frequency domain (MGate). For FFN, we adopt the Gated-Dconv Feed-forward Network (GDFN) in Restormer [40].

First, we apply DCT transform on feature map $\mathbf{X}_{in}^c$ of $c$th channel via:

$$\mathbf{Z}_{dct}^{h,w,c} = \sum_{u=0}^{H-1} \sum_{v=0}^{W-1} \mathbf{X}_{in}^{u,v,c} \cdot \mathbf{B}_{h,w}^{u,v}, \qquad (1)$$

where $\mathbf{Z}_{dct}^c \in \mathbb{R}^{H \times W}$ is the DCT frequency tokens, H and W are height and width of $\mathbf{X}_{in}^c$, respectively. $\mathbf{B}_{h,w} \in \mathbb{R}^{H \times W}$ is the basis image of the corresponding DCT coefficient located in $\mathbf{Z}_{dct}^{h,w,c}$, and there are $H \times W$ basis images for $\mathbf{X}_{in}^c$. Given indices $h, w$:

$$\mathbf{B}_{h,w}^{u,v} = F(u)F(v)\cos(\frac{\pi h}{H}(u+\frac{1}{2}))\cos(\frac{\pi w}{W}(v+\frac{1}{2})) \qquad (2)$$
$$s.t.\ u \in \{0,1,...,H-1\}, v \in \{0,1,...,W-1\},$$

where $F(u) = \begin{cases} \frac{1}{\sqrt{2}} & u = 0 \\ 1 & u > 0 \end{cases}$, $F(v) = \begin{cases} \frac{1}{\sqrt{2}} & v = 0 \\ 1 & v > 0 \end{cases}$.

Next, we describe DCT-LN, Freq-LC and MGate, respectively.

*DCT-LN.* LN [1] has been widely adopted in computer vision tasks due to its ability of stabilizing the training process [3]. Given the feature in spatial domain $\mathbf{X}_{in} \in \mathbb{R}^{H \times W \times C}$, as shown in Fig. 5, we first apply DCT to obtain the frequency tokens $\mathbf{Z}_{dct} = \text{DCT}(\mathbf{X}_{in}) \in \mathbb{R}^{H \times W \times C}$. Then, the LN after DCT can be defined as:

$$\mathbf{Z}_{norm} = \frac{\mathbf{Z}_{dct} - \bar{\mathbf{Z}}_{dct}}{\dot{\mathbf{Z}}_{dct}} \times \gamma + \beta, \qquad (3)$$

where $\bar{\mathbf{Z}}_{dct} = \frac{1}{C}\sum_{c=1}^{C} \mathbf{Z}_{dct}^c$, and $\dot{\mathbf{Z}}_{dct} = \sqrt{\frac{1}{C}\sum_{c=1}^{C}(\mathbf{Z}_{dct}^c - \bar{\mathbf{Z}}_{dct})^2 + \epsilon}$ is the standard deviation of $\mathbf{Z}_{dct}$ along the channel dimension. $\gamma$ and $\beta$ are learnable parameters, and $\epsilon = 10^{-5}$. After DCT, The distribution of frequency tokens varies greatly. Large amount of information stored in low frequency, and less information stored in the rest frequency. Thus, we adopt LN to force frequency tokens to be distributed equally, which is important to stabilize the training process. It is worth mentioning that applying LN before DCT would be equivalent of applying convolution in frequency domain (resulting

**Table 2: Explanation of some symbols.**

| Sym | Explanation | Sym | | Explanation |
|-----|-------------|-----|---|-------------|
| H : | Height | N = | HW: | Resolution of tensor |
| W: | Width | $\hat{C}$ = | C/r: | Channel for SA |
| C : | Channel | n = | $b^2$ : | Resolution of window |
| b : | Window | m = | N/n: | Numbers of windows |
| r : | SA heads | | | |

from a simple calculation), which do not help in balancing the distribution of frequency tokens.

*Freq-LC.* Given the feature $\mathbf{X}_{norm} \in \mathbb{R}^{H \times W \times C}$, we first apply 1×1 convolutions to obtain frequency-wise cross-channel context, and then $3 \times 3$ depth-wise convolutions are employed to gather channel-wise frequency local context. Through this way, we acquire $\hat{\mathbf{Q}}$, $\hat{\mathbf{K}}$ and $\hat{\mathbf{V}} \in \mathbb{R}^{r \times \hat{C} \times N}$ representing queries, keys, and values, where N, $\hat{C}$ and r are indicated in Table 2. Noted that we use different channels to represent multi-head. Then we design a window partition method and split $\hat{\mathbf{Q}}$, $\hat{\mathbf{K}}$ and $\hat{\mathbf{V}}$ into non-overlapping windows with the window size of b×b, acquiring $\tilde{\mathbf{Q}}$, $\tilde{\mathbf{K}}$, and $\tilde{\mathbf{V}} \in \mathbb{R}^{m \times r \times \hat{C} \times n}$, where m and n are indicated in Table 2. As shown in Fig. 4, we perform Local SA on the channel axis of $\tilde{\mathbf{Q}}$, $\tilde{\mathbf{K}}$, and $\tilde{\mathbf{V}}$. From each window $i$, features $\tilde{\mathbf{Q}}_i$, $\tilde{\mathbf{K}}_i$, and $\tilde{\mathbf{V}}_i \in \mathbb{R}^{\hat{C} \times n}$ can be obtained via flattening and transposing operations. Next, we perform SA to generate a transposed-attention map $\tilde{\mathbf{A}}_i \in \mathbb{R}^{C \times C}$ for window $i$. The process can be defined as $\tilde{\mathbf{A}}_i = \text{Softmax}(\tilde{\mathbf{Q}}_i \cdot \tilde{\mathbf{K}}_i^\top / \alpha)$ and $\text{Attention}(\tilde{\mathbf{Q}}_i, \tilde{\mathbf{K}}_i, \tilde{\mathbf{V}}_i) = \tilde{\mathbf{A}}_i \cdot \tilde{\mathbf{V}}_i$.

The transposed-attention map for all windows can be written as $\tilde{\mathbf{A}} = \{\tilde{\mathbf{A}}_1, \tilde{\mathbf{A}}_2, ..., \tilde{\mathbf{A}}_m\}$, where $\tilde{\mathbf{A}} \in \mathbb{R}^{m \times r \times \hat{C} \times \hat{C}}$, and $\alpha$ is a trainable scaling parameter.

*MGate.* To emphasize on the frequency, and control which complementary features should flow forward combined with Freq-LC, we apply MGate through intra-window MLP shown in Fig. 4 while sharing parameters on the other axes:

$$\text{MGate}(\tilde{\mathbf{V}}_i) = \sigma(\text{Linear}(\tilde{\mathbf{V}}_i)), \qquad (4)$$

where $\sigma$ indicates GELU operation. The intra-window MGate operation achieves local frequency mixing via aggregating cross-frequency context. Due to the global properties of each token in the frequency domain, it enhances the global information learned by Freq-LC from a different point of view. By combining Freq-LC and MGate branches together via element-wise multiplication, LoFT block can achieve superior performance to other counterparts.

After combining the output from Freq-LC and MGate by dot product, we perform window reverse to transpose the feature back to size $H \times W \times C$, and apply $1 \times 1$ convolutions to fuse cross-channel context indicated as $\mathbf{Z}_{axis}$. Correspondingly, we perform inverse DCT transform on feature map $\mathbf{Z}_{axis}$, whose feature on the $c$th channel is $\mathbf{Z}_{axis}^c$:

$$\mathbf{X}_{idct}^c = \sum_{h=0}^{H-1} \sum_{w=0}^{W-1} \mathbf{Z}_{axis}^{h,w,c} \cdot \mathbf{B}_{h,w}, \qquad (5)$$

where $\mathbf{B}_{h,w} \in \mathbb{R}^{H \times W}$ is the basis image for the corresponding DCT coefficient, $\mathbf{X}_{idct}^c \in \mathbb{R}^{H \times W}$ is the feature on the $c$th channel of $\mathbf{X}_{idct}$.

**Table 3: Comparison of different SAs. Spa, Freq, LS, GC, and LC mean Spatial, Frequency, Local Spatial-wise SA, Global Channel-wise SA, and Local Channel-wise SA, respectively. Spa Filter-GC means performing pass filters of different frequencies on the spatial feature, and then aggregating all global information. Symbols in computation complexity are indicated in Table 2. FLOPs (M) are calculated based on** $H = W = 256, C = 32, b = 8, r = 1$.

| SA | Model | Computation Complexity | FLOPs |
|---|---|---|---|
| Spa-LS | Uformer | $2NCn$ | 268 |
| Spa-GC | Restormer | $2NC\hat{C}$ | 134 |
| Freq-GC | - | $2NC(\hat{C} + \log_2(N))$ | 201 |
| Spa Filter-GC | - | $2NC(N\hat{C} + \log_2(N))$ | $\gg 10^3$ |
| Freq-LC | LoFormer | $2NC(\hat{C} + \log_2(N))$ | 201 |

*Complexity analysis.* As shown in Table 3, our Freq-LC shares the same computation complexity of Convolution and Attention compared with Spa-GC in Restormer [40]. Furthermore, the computational complexity of DCT required by the proposed approach only increases a manageable $O(N\log_2(N))$ while providing a significant improvement in performance.

# 4 UNDERSTANDING SA IN THE FREQUENCY DOMAIN

## 4.1 Spa-GC is equivalent to Freq-GC

For better reading, Table 3 lists different SA methods. To understand the physical meaning of matrix multiplication in the frequency domain, we explore the relationship between Spa-GC and Freq-GC. We have the proposition below:

PROPOSITION 1. *Spa-GC:* $\boldsymbol{O} = Attention(\boldsymbol{Q}, K, \boldsymbol{V})$ *and Freq-GC:* $\hat{\boldsymbol{O}}_f = IDCT(\hat{\boldsymbol{O}}) = IDCT(Attention(\hat{\boldsymbol{Q}}, \hat{\boldsymbol{K}}, \hat{\boldsymbol{V}}))$ *are identical, without considering depth-wise convolutions and DCT-LN, where queries, keys, and values are* $\boldsymbol{Q}, \boldsymbol{K}, \boldsymbol{V}$ *in the spatial domain, and* $\hat{\boldsymbol{Q}}, \hat{\boldsymbol{K}}, \hat{\boldsymbol{V}}$ *in the frequency domain.*

*Proof for Proposition 1.* For simplicity, we elaborate the deviation in 2D matrices instead of 3D tensors, *e.g.*, simplifying the size of the features $\hat{C} \times H \times W$ to $\hat{C} \times N$, where $N = HW$, shown below. The output features represented in the spatial domain after performing global SA on queries, keys, and values on the spatial ($\mathbf{Q}, \mathbf{K}, \mathbf{V} \in \mathbb{R}^{\hat{C} \times N}$) and frequency ($\hat{\mathbf{Q}}, \hat{\mathbf{K}}, \hat{\mathbf{V}} \in \mathbb{R}^{\hat{C} \times N}$) domains can be obtained via:

$$\mathbf{O} = \text{Softmax}(\mathbf{Q} \cdot \mathbf{K}^\top) \cdot \mathbf{V} \tag{6}$$

$$\begin{aligned}\hat{\mathbf{O}}_f = \text{IDCT}(\hat{\mathbf{O}}) &= \text{Softmax}(\hat{\mathbf{Q}} \cdot \hat{\mathbf{K}}^\top) \cdot \hat{\mathbf{V}} \cdot \mathbf{D} \\ &= \text{Softmax}(\mathbf{Q} \cdot \mathbf{D} \cdot \mathbf{D}^\top \cdot \mathbf{K}^\top) \cdot \mathbf{V} \cdot \mathbf{D} \cdot \mathbf{D}^\top \\ &= \text{Softmax}(\mathbf{Q} \cdot \mathbf{K}^\top) \cdot \mathbf{V} \\ &= \mathbf{O}, \end{aligned} \tag{7}$$

where $\mathbf{D} \in \mathbb{R}^{N \times N}$ is the matrix representation of DCT coefficients, and $\hat{\mathbf{O}}_f$ means the SA which is calculated in the frequency domain and then transformed back to the spatial domain via inverse DCT.

## 4.2 Analysis of Freq-LC from Spatial Perspective

We argue that our Freq-LC learns both coarse- and fine-grained global features, and explores different properties in representation. In this section, we analyze Freq-LC in the frequency domain from a new perspective.

As illustrated in Eq. 5, for an image with the size of $H \times W$, it can be represented as the sum of a series of basis images $\mathbf{B}_{h,w} \in \mathbb{R}^{H \times W}$ with the corresponding DCT coefficient, where $h \in \{0, ..., H-1\}, w \in \{0, ..., W-1\}$. We have the following proposition holds:

PROPOSITION 2. *Our Freq-LC can be seen as performing pass filters on a spatial feature, by representing the frequency tokens within a specific window as the summation of their corresponding basis images in the spatial domain. Compared to Freq-LC which applies Local Channel-wise SA on specific frequency tokens, realizing Freq-LC in the spatial domain would result in a significant increase in both memory and computation complexity.*

*Proof for Proposition 2.* We design a window partition method (window size = $b \times b$) and obtain $\tilde{\mathbf{Q}}, \tilde{\mathbf{K}}$ and $\tilde{\mathbf{V}} \in \mathbb{R}^{m \times \hat{C} \times n}$, where $n = b^2$ and $m = N/n$ after the window partition on $\mathbf{Q}, \mathbf{K}$ and $\mathbf{V}$, respectively. From each window $i$, features $\tilde{\mathbf{Q}}_i, \tilde{\mathbf{K}}_i, \tilde{\mathbf{V}}_i \in \mathbb{R}^{\hat{C} \times n}$ can be obtained via flattening and transposing operations. Let's assume padding $\tilde{\mathbf{Q}}_i, \tilde{\mathbf{K}}_i$, and $\tilde{\mathbf{V}}_i$ to size $\hat{C} \times N$ with zeros, obtaining $\ddot{\mathbf{Q}}_i, \ddot{\mathbf{K}}_i$ and $\ddot{\mathbf{V}}_i$. $\ddot{\mathbf{Q}}_i, \ddot{\mathbf{K}}_i$ and $\ddot{\mathbf{V}}_i$ can be regarded as the frequency spectrum after applying corresponding pass filters on features in the spatial domain. The output frequency features after applying SA for the $i$th window on $\tilde{\mathbf{Q}}_i, \tilde{\mathbf{K}}_i$, and $\tilde{\mathbf{V}}_i$ before padding and on $\ddot{\mathbf{Q}}_i, \ddot{\mathbf{K}}_i$ and $\ddot{\mathbf{V}}_i$ after padding can be calculated as:

$$\tilde{\mathbf{O}}_i = \text{Softmax}(\tilde{\mathbf{Q}}_i \cdot \tilde{\mathbf{K}}_i^\top) \cdot \tilde{\mathbf{V}}_i \tag{8}$$

$$\ddot{\mathbf{O}}_i = \text{Softmax}(\ddot{\mathbf{Q}}_i \cdot \ddot{\mathbf{K}}_i^\top) \cdot \ddot{\mathbf{V}}_i, \tag{9}$$

Noted that $\tilde{\mathbf{Q}}_i \cdot \tilde{\mathbf{K}}_i^\top = \ddot{\mathbf{Q}}_i \cdot \ddot{\mathbf{K}}_i^\top$. Thus, after window reverse ($wr$) operation, the total output of local SA is aggregated by:

$$\begin{aligned}\ddot{\mathbf{O}} = \sum_{i=1}^{m} \ddot{\mathbf{O}}_i &= \sum_{i=1}^{m} \text{Softmax}(\ddot{\mathbf{Q}}_i \cdot \ddot{\mathbf{K}}_i^\top) \cdot \ddot{\mathbf{V}}_i \\ &= \sum_{i=1}^{m} \text{Softmax}(\tilde{\mathbf{Q}}_i \cdot \tilde{\mathbf{K}}_i^\top) \cdot \ddot{\mathbf{V}}_i \end{aligned} \tag{10}$$

$$\begin{aligned}\tilde{\mathbf{O}}_w = wr(\tilde{\mathbf{O}}_i) &= wr(\text{Softmax}(\tilde{\mathbf{Q}}_i \cdot \tilde{\mathbf{K}}_i^\top) \cdot \tilde{\mathbf{V}}_i) \\ &= \sum_{i=1}^{m} \text{Softmax}(\tilde{\mathbf{Q}}_i \cdot \tilde{\mathbf{K}}_i^\top) \cdot \ddot{\mathbf{V}}_i \\ &= \ddot{\mathbf{O}}, \end{aligned} \tag{11}$$

where $\tilde{\mathbf{O}}_w$ means the SA calculated in local windows and then window reversing to obtain the total output.

Although the above two operations lead to identical results in terms of accuracy, they have different efficiency. We summarize the computation complexity of different SAs in Table 3, including SAs used in popular methods such as Uformer and Restormer. Our Freq-LC (see Freq-LC (LoFormer) in Table 3) is much more efficient compared with aggregating global SAs in spatial (see Spa Filter-GC in Table 3).

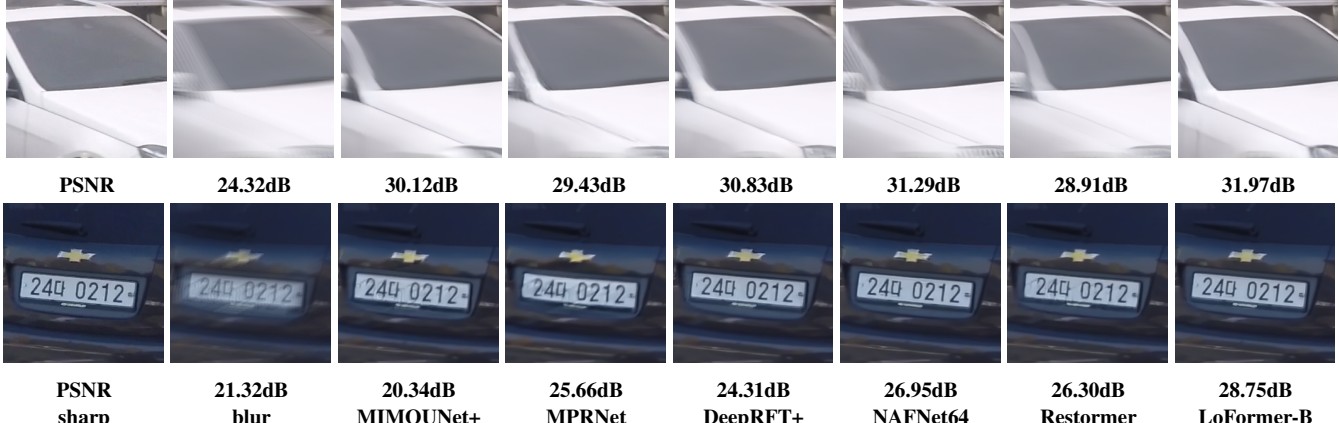

| PSNR | 24.32dB | 30.12dB | 29.43dB | 30.83dB | 31.29dB | 28.91dB | 31.97dB |

| PSNR | 21.32dB | 20.34dB | 25.66dB | 24.31dB | 26.95dB | 26.30dB | 28.75dB |
| sharp | blur | MIMOUNet+ | MPRNet | DeepRFT+ | NAFNet64 | Restormer | LoFormer-B |

**Figure 6: Examples on the GoPro test dataset. LoFormer produces better result.**

**Table 4: Summary of five public datasets.**

| Task | Dataset | Train | Val | Test | Types |
|---|---|---|---|---|---|
| | GoPro [20] | 2,103 | - | 1,111 | synthetic |
| | HIDE [28] | - | - | 2,025 | synthetic |
| Motion Deblur | RealBlur-R [25] | 3,758 | - | 980 | real-world |
| | RealBlur-J [25] | 3,758 | - | 980 | real-world |
| | REDS [21] | 24,000 | 3,000 | 300 | synthetic |

**Table 5: Comparison on GoPro [20], HIDE [28] datasets for setting $\mathcal{A}$. Transformer-based methods are highlighted in blue.**

| | GoPro | | HIDE | | Params | FLOPs |
|---|---|---|---|---|---|---|
| Method | PSNR | SSIM | PSNR | SSIM | M | G |
| DeepDeblur [20] | 29.08 | 0.914 | 25.73 | 0.874 | 11.7 | 336 |
| DeblurGAN [13] | 28.70 | 0.858 | 24.51 | 0.871 | - | - |
| DMPHN [42] | 31.20 | 0.940 | 29.09 | 0.924 | - | - |
| DBGAN [43] | 31.10 | 0.942 | 28.94 | 0.915 | 11.6 | 760 |
| MT-RNN [22] | 31.15 | 0.945 | 29.15 | 0.918 | - | - |
| MPRNet [41] | 32.66 | 0.959 | 30.96 | 0.939 | 20.1 | 588 |
| HINet [4] | 32.71 | 0.959 | 30.32 | 0.932 | 88.7 | 171 |
| MIMO-UNet+ [6] | 32.45 | 0.957 | 29.99 | 0.930 | 16.1 | 154 |
| NAFNet64 [3] | 33.69 | 0.967 | 31.32 | 0.943 | 65.0 | 64 |
| DeepRFT+ [18] | 33.52 | 0.965 | 31.66 | 0.946 | 23.0 | 187 |
| UFPNet [10] | 34.06 | 0.968 | 31.74 | 0.947 | 80.3 | 243 |
| Uformer [34] | 33.06 | 0.967 | 30.90 | 0.953 | 50.9 | 90 |
| Restormer [40] | 32.92 | 0.961 | 31.22 | 0.942 | 26.1 | 135 |
| Stripformer [31] | 33.08 | 0.962 | 31.03 | 0.940 | 20.0 | 170 |
| LoFormer-S | 33.73 | 0.966 | 31.51 | 0.946 | 16.4 | 47 |
| LoFormer-B | 33.99 | 0.968 | 31.71 | 0.948 | 27.9 | 73 |
| LoFormer-L | **34.09** | 0.969 | **31.86** | 0.949 | 49.0 | 126 |

## 5 EXPERIMENT

### 5.1 Experimental Setup

*Dataset.* We evaluate our method on the five datasets summarized in Table 4. Since existing methods adopt different experimental settings, we summarize them and report three groups of results:

$\mathcal{A}$. train on GoPro, and test on GoPro / HIDE respectively;
$\mathcal{B}$. train and test on RealBlur-J / RealBlur-R respectively;
$\mathcal{C}$. train and test on REDS dataset (follow HINet [4]).

*Implementation Details.* We adopt the training strategy used in Restormer [40] unless otherwise specified. *I.e.*, the network training hyperparameters (and the default values we use) are learning strategy (progressive learning), data augmentation (horizontal and vertical flips), training iterations (600k), optimizer AdamW ($\beta_1 = 0.9$, $\beta_2 = 0.999$, weight decay $1\times10^{-4}$), initial learning rate ($3\times10^{-4}$). The learning rate is steadily decreased to $1\times10^{-6}$ using the cosine annealing strategy [17]. For LoFormer-S and LoFormer-B, we start training with patch size 128×128 and batch size 64. The patch size and the batch size pairs are updated to [(160, 40), (192, 32), (256, 16), (320, 8), (384, 8)] at iterations [184K, 312K, 408, 480K, 552K]. Due to statistics distribution shifts between training and testing [7], we utilize a step of 352 to perform 384×384 size sliding window with an overlap-size of 32 for testing. We set b = 8 for LoFormer-S and LoFormer-B. In the selection of the loss function, two kinds of loss functions are utilized: (1) L1 loss: $\mathcal{L}_1 = ||\hat{S} - S||_1$, and (2) Frequency Reconstruction (FR) loss [6, 18, 32]: $\mathcal{L}_{fr} = ||\mathcal{F}(\hat{S}) - \mathcal{F}(S)||_1$. Where $\hat{S}$, $S$ and $\mathcal{F}(\cdot)$ represent the predicted sharp image, the groundtruth sharp image and 2D Fast Fourier Transform, respectively. For LoFormer, the loss function $\mathcal{L} = \mathcal{L}_1 + 0.01\mathcal{L}_{fr}$.

*Evaluation metric.* Performances in terms of PSNR and SSIM over all testing sets, as well as the number of parameters and FLOPs are calculated using official algorithms.

### 5.2 Main Results

*Setting $\mathcal{A}$.* For setting $\mathcal{A}$, we train our model on 2,103 image pairs from GoPro [20], and compare them with several SOTA methods through the test set of GoPro [20] and HIDE [28]. As shown in Table 5 and Fig. 6, LoFormer outperforms the other CNN-based,

**Table 6: Comparison on RealBlur [25] dataset for setting $\mathcal{B}$.**

| Method | RealBlur-R PSNR | RealBlur-R SSIM | RealBlur-J PSNR | RealBlur-J SSIM | Params M | FLOPs G |
|---|---|---|---|---|---|---|
| DeblurGAN-v2 [14] | 36.44 | 0.935 | 29.69 | 0.870 | - | - |
| SRN [30] | 38.65 | 0.965 | 31.38 | 0.909 | - | - |
| MPRNet [41] | 39.31 | 0.972 | 31.76 | 0.922 | 20.1 | 777 |
| MAXIM [32] | 39.45 | 0.962 | 32.84 | 0.935 | 22.2 | 339 |
| DeepRFT+ [18] | 40.01 | 0.973 | 32.63 | 0.933 | 23.0 | 187 |
| Stripformer [31] | 39.84 | 0.974 | 32.48 | 0.929 | 20.0 | 170 |
| FFTformer [12] | 40.11 | 0.975 | 32.62 | 0.933 | 16.6 | 132 |
| LoFormer-B | **40.23** | 0.974 | **32.90** | 0.933 | 27.9 | 73 |

**Table 7: Comparison on the REDS-val-300 from REDS [21] dataset of NTIRE 2021 Image Deblurring Challenge Track 2 JPEG artifacts for setting $\mathcal{C}$.**

| Model | PSNR | SSIM | FLOPs | Params |
|---|---|---|---|---|
| MPRNet | 28.79 | 0.811 | 777 | 20.1 |
| HINet | 28.83 | 0.862 | 171 | 88.7 |
| MAXIM | 28.93 | 0.865 | 339 | 22.2 |
| NAFNet64 | 29.09 | 0.867 | 64 | 65.0 |
| LoFormer-B | **29.20** | **0.869** | 73 | 27.9 |

Transformer-based and MLP-based methods in both PSNR and SSIM on the GoPro test set. Under the same training strategy, our LoFormer-L achieves 1.17 dB gain over Restormer [40] with similar FLOPs (126G *vs.* 135G). Besides, LoFormer-L obtains robust results on other datasets. For HIDE test set, LoFormer-L achieves 31.86dB, 0.64dB higher than Restormer. Note that as mentioned in the introduction, Spa-LS in Uformer hurts long-range modeling, Spa-SS in Stripformer relies on a strong assumption, and Spa-GC in Restormer suffers from fine-grained correlation deficiency, while our Freq-LC consists of simple yet effective operations to model long-range dependency without losing fine-grained details. More comparisons between Freq-LC and Spa-GC are analyzed in depth in the introduction.

*Setting $\mathcal{B}$.*. As can be seen in Table 6, LoFormer-B achieves 32.90dB on RealBlur-J test set, 0.42dB higher than Stripformer [31] with fewer FLOPs (73G *vs.* 170G). For RealBlur-R, LoFormer-B also gets a better outcome (40.23dB) than Stripformer (39.84dB).

*Setting $\mathcal{C}$.*. Moreover, LoFormer-B achieves a competitive result with other methods for REDS [21] dataset shown Table 7, *e.g.*, 0.27dB better than MAXIM. In brief, the quantitative experimental results indicate that our LoFormer has a good ability to handle motion deblurring tasks under different conditions.

## 5.3 Analysis and Discussion

Extensive ablation studies are conducted to verify the effectiveness of LoFT block *w.r.t.* different components. The models are

**Table 8: Ablation studies. LN-DCT: LayerNorm followed by DCT. DC: Dilated Channel-wise SA with a dilated stride of $[H/b, W/b]$. CGate: applying Linear on the channel axis.**

| Attention | LN-DCT | DCT-LN | MGate | PSNR | FLOPs |
|---|---|---|---|---|---|
| Spa-GC | × | × | × | 32.74 | 43.33 |
| Freq-GC | ✓ | × | × | 32.68 | 44.46 |
| | × | ✓ | × | 32.84 | 44.46 |
| Freq-DC | ✓ | × | × | 32.75 | 44.46 |
| | × | ✓ | × | 32.90 | 44.46 |
| Freq-LS | ✓ | × | × | 32.95 | 45.90 |
| | × | ✓ | × | 33.15 | 45.90 |
| Freq-LC | ✓ | × | × | 32.91 | 44.46 |
| | ✓ | × | ✓ | 32.94 | 46.97 |
| | × | ✓ | × | 33.17 | 44.46 |
| | × | ✓ | CGate | 32.97 | 47.95 |
| | × | ✓ | ✓ | **33.23** | 46.97 |
| LoFormer-B | × | ✓ | × | 33.54 | 69.68 |
| | × | ✓ | ✓ | **33.99** | 73.04 |

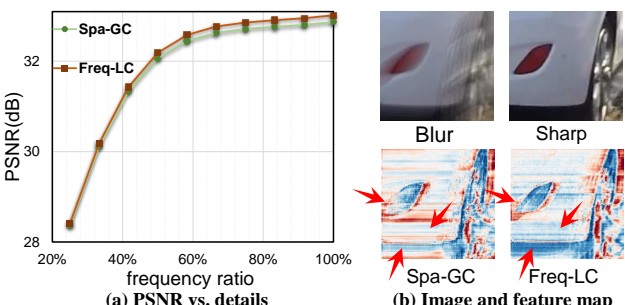

(a) PSNR vs. details  (b) Image and feature map

**Figure 7: (a) PSNR vs. details on GoPro. The higher ratio means in testing, more high-frequency information (details) of the restored images (like Fig. S2) is included when calculating PSNR. Freq-LC performs better than Spa-GC on details. (b) Feature maps of Spa-GC and Freq-LC. Freq-LC gets better details than Spa-LC (red arrows).**

trained on GoPro with progressive learning strategy for 300K iterations. The training starts with patch size 128×128 and batch size 32, which shares the same hyper-parameters of the model design with LoFormer-S.

## 5.4 Effectiveness of DCT-LN

We propose to conduct Layer Normalization (LN) on the frequency-transformed matrix $\mathbf{X}_{dct}$ rather than the input matrix $\mathbf{X}_{in}$, aiming to ensure an equitable distribution of frequency tokens, thereby promoting training stability. As demonstrated in Table 8, the utilization of DCT-LN alongside Freq-LC results in superior performance, yielding a 0.26 dB improvement compared to employing LN-DCT (32.91 dB). Similarly, for Freq-GC, employing DCT-LN contributes to a gain of 0.16 dB.

## 5.5 Effectiveness of Local Attention

In the frequency domain, we can easily acquire global information of the input feature $\mathbf{X}_{in}$ within a local window of $\mathbf{X}_{dct}$. Compared with

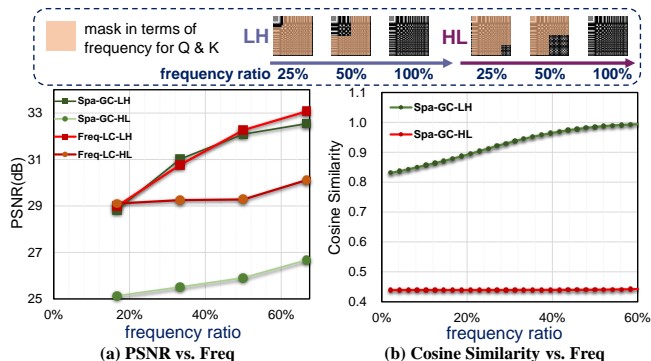

**(a) PSNR vs. Freq**

**(b) Cosine Similarity vs. Freq**

**Figure 8: The analysis for Spa-GC and Freq-LC on GoPro dataset. (a) PSNR *vs.* proportions of Low-frequency (LF) or High-frequency (HF); (b) The cosine similarity of Spa-GC between the source attention map calculated through different ratios of LF/HF and the target attention map calculated through all the information. A detailed illustration on the horizontal axis is shown in supplementary.**

Freq-GC (32.68dB) in Table 8, Freq-LC acquires a gain of 0.23 dB in PSNR (32.91dB). Fig. 7 (a) shows masking different detail ratios of restored images when computing PSNRs. Freq-LC is more capable of capturing **high-frequency details**, which are suppressed by structure features in popular Spa-GC (see Fig. 8(a)). In addition, Fig. 7 (b) further indicates that Freq-LC acquires better feature details than Spa-GC for deblurring. Freq-LC performs similarly to Spa-GC in low-freq, but it achieves better results when involving more high-freq, showing Freq-LC restores details better than Spa-GC. Moreover, we design a Dilated Channel-wise SA in the frequency domain (Freq-DC) to further verify the effectiveness of learning coarse-and-fine information separately in SA. Even applying DCT-LN to make frequency tokens to be distributed equally, Freq-DC (32.90dB) performs 0.27dB worse than Freq-LC (33.17dB).

## 5.6 Effectiveness of MGate

Table 8 shows that MGate boosts the effectiveness of Freq-LC in linear time, and helps Freq-LC filter out the invalid information, which let LoFormer-B develop deeper (33.99dB *w/* MGate *vs.* 33.54dB *w/o* MGate). Additionally, we perform MLP on the channel axis of $\mathbf{X}_{dct}$, named as CGate. The performance drops compared with MGate (32.97dB *vs.* 33.23dB in Table 8), showing the complementary feature provided via MGate operation.

## 5.7 Discussion

***Spatial or Channel***. As delineated in Table 8, the efficacy of Freq-LS (33.15 dB, 45.90G) exhibits a comparative level to that of Freq-LC (33.17 dB, 44.46G), with the former incurring slightly elevated computational complexity. Consequently, we opt to integrate Freq-LC into the LoFormer architecture.

***Superiority of Freq-LC***. To demonstrate the superior ability of Freq-LC over Spa-GC in learning fine-grained high-frequency features, a comparison is made between the two methods by plotting PSNR curves against low-/high-frequency ratio on Fig. 8 (a). The purpose of this analysis is to highlight the advantage of Freq-LC in

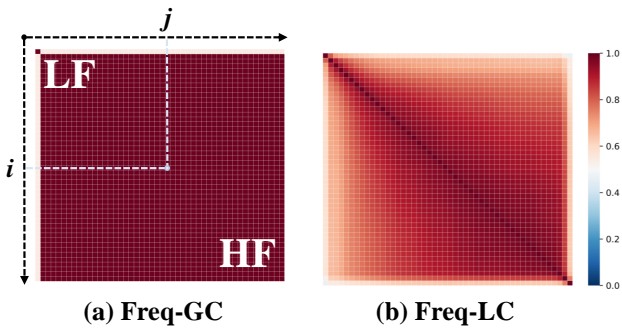

**(a) Freq-GC**  **(b) Freq-LC**

**Figure 9: Cross cosine similarity among attention maps of different windows along the diagonal line in Stage-1. For example, the value in position $(i, j)$ means the cosine similarity between $\tilde{\mathbf{A}}^{i,i}$ and $\tilde{\mathbf{A}}^{j,j}$, where $i \in \mathbb{R}^{H/b}$, $j \in \mathbb{R}^{W/b}$ and $\tilde{\mathbf{A}} \in \mathbb{R}^{H/b \times W/b \times C \times C}$.**

capturing subtle details that may be suppressed by structure features in Spa-GC. We test LoFormer framework with Freq-LC and Spa-GC, but with different selected frequency ratios. Our Freq-LC observes a performance boost at a faster rate (see Freq-LC-LF and Spa-GC-LF) when involving more and more high-frequency parts. Conversely, involving more low-frequency parts does not change the performance of Freq-LC dramatically like Spa-GC (see Freq-LC-HF and Spa-GC-HF), which proves that high-frequency information plays a more important role in Freq-LC than Spa-GC. Similarly, Fig. 8 (b) shows Spa-GC does not effectively learn high-frequency information, whose cosine similarity with the attention map of Spa-GC is small (see Spa-GC-HF). Besides, computing SA within different frequency windows helps to explore divergent properties in representation.

***Attention Maps***. To better understand whether the attention maps learned from each local window are the same or different, we calculate the Cosine Similarity matrices of the attention maps in stage-1. As shown in Fig. 9, the attention maps in Freq-GC (32.84dB) are similar to each other, which suppresses the network's learning ability for high-frequency (fine) information. While the attention maps in Freq-LC (33.17dB) are quite different from each other, indicating that different independent local windows provide information in different ways for image deblurring.

## 6 CONCLUSIONS

We introduce a novel approach termed Local Frequency Transformer (LoFormer) for image deblurring. In contrast to prior transformer-based methodologies that focus on either learning localized self-attention (SA) mechanisms or adopting coarse-grained global SA strategies to mitigate computational complexity, LoFormer offers a unique solution. It simultaneously captures both coarse- and fine-grained long-range dependencies by employing channel-wise self-attention within localized frequency token windows. Moreover, we incorporate MLP Gating to augment global learning capabilities and eliminate irrelevant features. Extensive experiments across five image deblurring datasets demonstrates the superior performance of our proposed LoFormer.

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
