# OpenReview forum: "LoFormer: Local Frequency Transformer for Image Deblurring"
_acmmm.org/ACMMM/2024/Conference — MM2024 Poster_

### Official Review · Reviewer_eXy8 · 2024-05-15

**Rating:** 3
**Confidence:** 4

**Summary:**

The paper discusses the development of a novel approach for image deblurring called the Local Frequency Transformer (LoFormer). This method incorporates a Local Channel-wise Self-Attention (SA) in the frequency domain to capture cross-covariance within low- and high-frequency local windows, aiming to address the computational challenges associated with traditional SA techniques. It enhances the ability to model long-range dependencies without losing detail and expands the range of representational properties. Additionally, a complementary MLP Gating mechanism is introduced to filter out irrelevant features and boost global learning capabilities, which significantly improves performance on the GoPro dataset.

**Strengths:**

- LoFormer provides with a decent experimental result compared to SOTA method with similar # params.
- The paper is well written, easy to catch authors intention, and effectively structured.
- I like the Figure 4, which is very clear to explain your methods.

**Limitations:**

- I used to try DCT/iDCT in other fields, the main drawbacks of this transformation is highly computation cost. As illustrated in Table 5, compared with same level of # params, the FLOPs value of your method is approximately 40% higher than that of other SOTA methods (eg, Uformer), which may have contributed significantly to the substantial improvements in PSNR/SSIM. This could significantly undermine the motivation and contribution of your method, as your purpose is to achieve the best results within a given limited computational complexity.
- The computational complexity of your method for generating the DCT is stated as #O(N \log_2(N))#, which is factually incorrect; it should be #O(N^2 \log_2(N))#. Additionally, your method also requires performing an iDCT, which also has a complexity of #O(N^2 \log_2(N))#. Therefore, the actual increase in complexity of your method should be #2O(N^2 \log_2(N))#. This should be considered a major issue in your paper, necessitating further explanation during the rebuttal phase.
- Section 4.2 is somewhat unclear. Your theoretical analysis should demonstrate that your method's performance and computational complexity bounds are superior to those of others, necessitating further elucidation through additional formulaic analysis.

**Suitability:**

3

---

### Official Review · Reviewer_xc9w · 2024-05-20

**Rating:** 5
**Confidence:** 4

**Summary:**

1.	This paper proposes a novel approach termed Local Frequency Transformer (LoFormer) for image deblurring, which achieves superior global modeling and fine-grained feature correlation.
2.	The motivation of the paper is clear, the regulations are clear, the experiments are sufficient, and the formatting is excellent.

**Strengths:**

See Summary

**Limitations:**

1.	Please elaborate further on Eq. 7.
2.	Some image restoration methods also perform deblurring tasks [1,2], so please compare their methods.
3.	Please cite the related literature [3].
4.	Please provide the inference time comparisons with some mainstream methods.
5.	In Figure 7 (b), which color bar you choose to plot the feature maps?

[1] Cui, Yuning, et al. "Focal network for image restoration." Proceedings of the IEEE/CVF international conference on computer vision. 2023.

[2] Cui, Yuning, et al. "Selective frequency network for image restoration." The Eleventh International Conference on Learning Representations. 2022.

[3] Dong, Shuting, et al. "Enhanced Image Deblurring: An Efficient Frequency Exploitation and Preservation Network." Proceedings of the 31st ACM International Conference on Multimedia. 2023.

**Suitability:**

3

---

### Official Review · Reviewer_do2s · 2024-05-22

**Rating:** 5
**Confidence:** 4

**Summary:**

This paper proposes the Local Frequency Transformer with novel Freq-LC. The motivation of the paper is clear and the experiment is sufficient.

**Strengths:**

1. Innovation is enough for me.

2. Experiments prove the effectiveness of each module.

3. Symbol refinement is beneficial to the paper.

**Limitations:**

1. In Fig 7, more feature maps need to be provided.

2. Is Loformer only compatible with deblurring tasks? If it is generic, please add 1-2 tasks (e.g. defocusing deblurring, deraining)

3. More works in relation to frequency domain should be discussed.

    [1] Learning Frequency Domain Priors for Image Demoireing.

    [2] Feature Modulation Transformer: Cross-Refinement of Global Representation via High-Frequency Prior for Image Super-Resolution

    [3] Learning Frequency-aware Dynamic Network for Efficient Super-Resolution

**Suitability:**

3

---

### Official Review · Reviewer_mBg3 · 2024-05-24

**Rating:** 2
**Confidence:** 4

**Summary:**

The paper generalizes frequency in the image deblurring domain and proposes LoFormer which applies Local Channel-wise Self-Attention in the frequency domain and can capture both coarse and fine-grained long-range dependencies.
It argues that existing methods of SA either suffer from fine-grained correlation deficiency or have quadratic computational complexity. This paper utilizes a Local Frequency Transformer (LoFT) block, which applies Local Channel-wise Self-Attention in the frequency domain, ensuring equitable learning opportunities for structural and detailed features.
Additionally, an MLP Gating mechanism is incorporated to filter irrelevant features and enhance global learning. Experiments demonstrate that LoFormer outperforms other state-of-the-art methods.

**Strengths:**

1. LoFormer's use of Local Channel-wise Self-Attention in the frequency domain allows efficient modeling of long-range dependencies without sacrificing fine details.
2. LoFormer is much more efficient compared with aggregating global SAs in spatial.
3. The paper provides sufficient theoretical proof that Spa-GC equals Freq-GC, where coarse information dominates the calculation.

**Limitations:**

1.	In the Method: Local Frequency Transformer Block section, the description of Freq-LC is too concise, and since this section is supposed to be the core of the paper, there is no formula to describe the process.
2.	There is no explanation in the text about the Latent module in Figure 4. It’s hard to understand the specific relationship between the Latent module and other modules only by the presentation of Figure 4.
3.	While the paper claims LoFormer has a manageable computational complexity, the use of frequency transformations and SA could still be computationally intensive compared to simpler models, especially for very high-resolution images. A comparison of training and inference times is missing.
4.	As illustrated in the paper, the model is quite complex which might have challenges in implementation and may require specialized hardware or software optimizations for efficient training and inference.

**Suitability:**

3

---

### Meta-Review · Area_Chair_Vry2 · 2024-07-01

**Recommendation:** Accept (Poster)
**Confidence:** 5

**Metareview:**

This paper was reviewed by four experts in the field, with (1) borderline reject, (1) accept, (1) weak accept and (1) borderline accept. It proposed a local frequency transformer with the novel Freq-LC for the image deblurring, which is well motivated. The design is novel with frequency domain attention for small model and faster inference. The experimental results are well organized to support the conclusion. Based on the reviewers' feedback, the decision was made to recommend it for acceptance to ACMMM 2024. We congratulate the authors on their acceptance.